# Synthesis of the Titanium Oxides Using a New Microwave Discharge Method

**DOI:** 10.3390/ijms26052173

**Published:** 2025-02-28

**Authors:** Marian Mogildea, George Mogildea, Sorin I. Zgura, Gabriel Chiritoi, Cristian Ionescu, Valentin Craciun, Petronela Prepelita, Natalia Mihailescu, Alexandru Paraschiv, Bogdan Stefan Vasile, Catalin Daniel Constantinescu

**Affiliations:** 1Institute of Space Science—Subsidiary of the National Institute for Laser, Plasma and Radiation Physics, 077125 Magurele, Romania; marian_mogildea@spacescience.ro (M.M.); szgura@spacescience.ro (S.I.Z.); gabriel.chiritoi@spacescience.ro (G.C.); idcristi@gmail.com (C.I.); 2National Institute for Laser, Plasma and Radiation Physics, 409 Atomistilor st., 077125 Magurele, Romania; valentin.craciun@inflpr.ro (V.C.); petronela.garoi01@gmail.com (P.P.); natalia.serban@inflpr.ro (N.M.); 3Extreme Light Infrastructure for Nuclear Physics, 077125 Magurele, Romania; 4Romanian Research & Development Institute for Gas Turbines, 061126 Bucharest, Romania; alexandru.paraschiv@comoti.ro; 5National Research Center for Micro and Nanomaterials, Bucharest National Polytechnic University of Science and Technology, 060042 Bucharest, Romania; vasile_bogdan_stefan@yahoo.com; 6Research Center for Advanced Materials, Products and Processes, Bucharest National Polytechnic University of Science and Technology, 060042 Bucharest, Romania; 7CNRS, LP3 UMR 7341, Aix-Marseille University, 13009 Marseille, France; catalin.constantinescu@cnrs.fr

**Keywords:** microwaves, plasma, nanoparticles

## Abstract

This research highlights the different behaviors of titanium (Ti) wires under the action of 500 W and 800 W microwave power levels. Following the interaction between microwaves and a titanium wire placed in the node of the (TM_011_—transverse magnetic mode) waveguide in air at atmospheric pressure, plasma was generated. Using optical emission spectroscopy technique it was observed that during plasma generation at 500 W and 800 W microwaves powers, metallic ions, and gas ions were created, and the plasmas fulfilled the local thermodynamic equilibrium (LTE) conditions. The XRD analysis showed that on the surface of the Ti wire exposed to 500 W microwave power a mixture of titanium dioxide (TiO_2_) and titanium oxide (TiO) grew, while the Ti wire exposed to 800 W microwave power was completely vaporized and a mixture of TiO_2_ and TiO nanoparticles was synthesized. The SEM analysis showed that the dimensions of the titanium oxide (TiO_x_) nanoparticles generated by both microwave discharges ranged from 5 nm to 200 nm. The results of EDS analysis showed that the power of microwaves plays an important role in quantitative conversion from Ti wire into a TiO_x_ mixture. The TEM analysis indicates that most of the nanoparticles are either amorphous or nanocrystalline. Using this simple and inexpensive technique one can grow a TiO_x_ layer on the surface of titanium electrodes or can synthetize nanocrystalline TiO_x_ particles.

## 1. Introduction

Titanium oxides (TiO_x_) are naturally occurring compounds used for many years in various branches of industry [1]. TiO_x_ is a non-toxic substance that possesses antibacterial properties [2,3] and has excellent biocompatibility with organic cells [3,4]. TiO_x_ in powder form [5] is used in commercial applications as pigments, in the food industry as a coloring additive, and in the pharmaceutic domain as a protector of active ingredients against light radiation [6]. The deposition of thin films or coatings on the surfaces of other materials with TiO_2_ allows for the development of new materials with interesting physical and chemical properties. Until now, TiO_2_ and TiO thin films were successfully synthesized by chemical and physical techniques such as chemical bath deposition (CBD) [7], chemical vapor deposition (CVD) [8], molecular beam epitaxy (MBE) [9], RF magnetron sputtering, electrodeposition [10], and pulsed laser deposition (PLD) [11].

Current technologies based on plasma generation have low efficiency and incomplete conversion of the titanium to TiO_2_ [12,13]. Moreover, these systems contain mechanical and electronic modules that are very expensive, have large dimensions, and cannot be easily installed in another workspace. The chemical techniques are less expensive, but they have limitations in terms of mass manufacturing, special condition requirements, as well as the use of toxic or poisonous chemicals that endanger human health and the environment [14]. In addition, after the manufacture of the nanoparticles, it is necessary to implement procedures for washing and drying the nanoparticles. These procedures require time and money spent, which leads to a higher cost of producing nanoparticles. Considering that the demand for TiO_x_ nanoparticles and new materials based on TiO_x_ is increasing, our study investigated the growth of TiO_x_ directly on the surface of titanium wires and the synthesis of TiO_x_ nanoparticles using a new microwave plasma method. The interaction processes of microwaves with dielectric materials and metals are very well documented. It is known that microwaves are reflected by bulk metals and are absorbed by dielectric materials (gaseous, liquids, plastic, and ceramic materials). Following the absorption process of microwaves by dielectric materials, they are heated up and melted, and for very large powers a plasma is created [15]. Recently, studies were performed on the heating of metal powders under the action of microwaves. The results showed that the physical processes occurring during the interaction between microwaves and metal particles are of an electrical nature. When the metallic particles are mixed with liquid substances and are exposed to the microwave field, the liquids heat up due to the occurrence of electrical discharges between the particles [16]. Or, when the metal powder is in a gaseous atmosphere and is exposed to microwaves, it heats up through the occurrence of Eddy currents [17]. Currently, we have a good understanding of the physical processes occurring during the interaction between small metallic objects (feature size < 1 mm) and microwaves. Studies have been carried out on the interaction of microwaves and electrons in vacuum and gaseous atmospheres. Research carried out in a vacuum has shown that when a metal electrode is exposed to a microwave field, electrons are generated through the field emission process [18]. Other experiments showed that when the electrodes of a photomultiplier detector interact with microwaves an electrical signal at the output of the photomultiplier is generated [19]. Popescu S. et al. [20] showed that in atmospheric air conditions, a plasma can be created between two titanium pieces brought into contact when these were irradiated with microwaves. In 2018, Yukun Feng et al. [21] introduced a quartz glass cylinder containing paper clips and gaseous acetone into a microwave oven. Following the interaction between the microwaves and the metallic paper clips, electric sparks were generated, and the acetone was decomposed. Other research performed in air or CO_2_ atmospheres has shown that when a metal electrode is exposed to a microwave field at a certain power level, plasma is generated and the metallic electrode is vaporized [22]. From these studies, it was observed that during the interaction of microwaves with metallic wires, two physical processes are involved: field emission and the thermionic emission process [23]. The field emission is responsible for plasma initiation and the thermionic process is responsible for the ion generation and vaporization of the metallic wire.

Therefore, the microwave plasma method described in this paper uses the same physical process to generate a plasma, which is oxidizing an electrically insulated titanium wire at low power and synthesizing titanium oxide nanoparticles at high power levels.

Considering that numerous studies have been carried out recently on the interaction of metal powders/wires or small metal objects with microwaves of a certain power, this work highlights different behaviors of Ti wires under the action of low and high microwave power levels. Moreover, using this inexpensive method one can obtain a high-efficiency conversion of the titanium in crystalline TO_x_ powder nanoparticles and grow the TO_x_ layers on the surface of the Ti electrodes in atmospheric air.

## 2. Results

To determine the plasma parameters, the plasma was investigated using the optical emission spectroscopy method. An Ocean Optics USB 2000 ++ spectrometer linked to an optical fiber (Ocean Optics Inc., Orlando, FL, USA) was used to record the optical spectrum of the plasma [24]. The integration time of the USB 2000 ++ software was set at 10 ms. Before starting the measurements, and the spectrometer was calibrated using a broadband light source (Ocean Optics DH-mini UV-Vis-NIR Deuterium—Halogen Light Source). After data acquisition, the emission spectrum of the plasma was compared with the NIST database [25]. The optical emission spectra of the plasma generated at 500 W and 800 W microwave power are displayed in Figure 1 and Figure 2, respectively. As one can observe, the emission lines of excited atoms and ions from the atmosphere (O_2_ and N_2_) and wire (Ti) are present in the acquired spectra.

To evaluate if the plasma is at local thermodynamic equilibrium (LTE), the electronic temperature of the plasma was determined using the Boltzmann method [26].

The plots for the plasmas generated using 500 W and 800 W microwave power are displayed in Figure 3 and Figure 4, respectively.

During plasma generation different behaviors of titanium wires depending on the microwave power were observed. For the Ti wire exposed to 500 W microwave power, a thick layer of Ti oxide was formed on the surface of the wire (see Figure 5), while for the Ti wire exposed to 800 W microwave power, this was vaporized (Figure 6) and Ti oxide was deposited on the walls of the waveguide and a silicon substrate placed inside the waveguide to collect the generated nanoparticles for further analysis.

In Figure 5a, an optical microscope image of the Ti wire after exposure at 500 W microwave power shows that the wire was oxidized along a length of 3 mm starting from the tip.

In Figure 6a,b images of the Ti wire before and after exposure to 800 W microwave power are displayed. Before starting the plasma generation process, the tip of the Ti wire was 5 mm long. After 10 s when the plasma generation process was stopped, the tip of the titanium wire was completely vaporized (see yellow zone in Figure 6b).

To determine the morphology and oxidation state of the Ti oxide synthetized during microwave discharge, the tip of the Ti wire exposed to 500 W microwave power and particles deposited on the silicon substrate obtained at 800 W microwave power were analyzed using EDS, SEM, TEM, and XRD. A high-resolution scanning electron microscope (HRSEM) was used for topographic analysis of samples as well as to observe the structural quality and surface morphology of samples. More precisely, the Apreo microscope from FEI (Thermo Fisher Scientific, Waltham, MA, USA) allows a 0.9 nm resolution. By using HRSEM, we studied the surface of the samples at different magnifications, in a high vacuum, by scanning them with a beam of accelerated electrons at very high energies (≈20 keV) from a 14 cm to 10 cm working distance and an electrical current from 0.1 nA to 13 pA. Prior to imaging analysis, a thin layer of gold was sputtered onto the sample surfaces to avoid electrostatic charging during measurements. For compositional analyses, the system was equipped with an X-ray source and an EDX unit with elementary energy dispersion spectroscopy (EDS), a fixed silicon detector, and an integrated Peltier element as a cooling system. For EDX, it was operated at 25 kV acceleration voltage, 3.2 nA electrical current, and the dead time was 34 s. EDX measurements were applied at 2.000× magnification. The quantitative elemental analyses were done using TEAM^TM^ v.4.5 software. Figure 7 displays the EDS spectrum acquired from the surface of the Ti wire after exposure to 500 W microwave power and Table 1 displays the chemical composition of the TiO_x_.

In Figure 8 the SEM image (60,000× magnification) of the tip of the Ti wire after exposure to 500 W microwave power is displayed. The sizes of nanoparticles from the surface of the Ti wire ranged between 5 nm and 200 nm.

Figure 9 displays the XRD patterns acquired from the surface of the Ti wire exposed to 500 W microwave power irradiation and their analysis using the Malvern Panalytical software package (https://www.malvernpanalytical.com/en/products/category/x-ray-diffractometers). The observed diffraction peaks correspond to TiO_2_ anatase and several TiO phases. No metallic Ti peaks were observed, an indication that the formed oxide layer was rather thick.

The EDS, SEM and XRD analyses performed on the nanoparticles collected on the silicon substrate during vaporization of the Ti wire at 800 W microwave power showed that they have dimensions between 5 nm and 200 nm and consist of a mixture of TiO_2_ and TiO. Figure 10 displays the EDS spectrum acquired from the nanoparticles deposited on the silicon substrate at 800 W microwave power and Table 2 displays the chemical composition of the TiO_x_.

Figure 11 displays an SEM image of the nanoparticles deposited on the silicon substrate from Ti wire irradiated at 800 W microwave power.

The XRD analysis (see Figure 12) of the nanoparticles collected on the Si substrate indicates that their composition is a mixture of TiO and TiO_2_. However, the diffraction peaks are smaller and wider than those obtained from the metallic wire irradiated at 500 W. Some TiO_2_–SiO_2_ mixture, formed because the plasma was very hot and interacted with the Si substrate, was also identified in the collected material.

Figure 13a,b show images of the Ti oxide nanoparticles deposited on the Si substrate using the transmission electron microscopy (TEM) technique. Figure 13c shows the selected area electron diffraction obtained from these particles, which indicates that most of the nanoparticles are either amorphous or nanocrystalline. In the high-resolution TEM image of the nanoparticles displayed in Figure 13b, some regions where interference fringes are visible can be observed, indicative of small crystalline regions.

After turning off the plasma generation, it was noticed that the titanium oxide nanoparticles deposited on the Si substrate at 800 W microwave power were very easy to brush off, while the oxide layer grown on the tip of the Ti wire exposed to 500 W microwave power had strong adhesion. To determine the adhesion of the Ti oxide formed on the tip of the Ti wire when this was exposed to 500 W microwave power, a Revetest Anton Paar instrument was used. The adhesion analyses of a film deposited on a substrate were estimated from the controlled scratching with a diamond-tipped needle (200 μm diameter of the needle tip) on the surface of the object. The test parameters were set at 2 mm/min advance with the depth force progressively increased between 0 N and 100 N.

In Figure 5b one can see the scratch performed by the needle of the instrument on the surface of the microwaves exposed Ti wire. During the adhesion testing, it was observed that when the needle of the instrument penetrated 0.2 mm into the sample, it exfoliated the oxide film from the wire. Figure 14 displays the graph of TiO_x_ adhesion versus pressing force on the surface of the titanium wire.

The analysis displayed in Figure 14 shows that the TiO_x_ layer was exfoliated at ~50 N pressing force, (at ~1 mm at the interface between metal and TiO_x_). After removing the oxide layer, it was noticed that it was very hard and brittle.

## 3. Discussion

Microwaves are defined in the electromagnetic spectrum as non-ionizing radiation because the photons do not have enough energy to ionize the substance. However, it has been observed that when microwaves interact directly with wires or small metallic objects in a gaseous atmosphere, ions are generated. The triggering factor in the plasma initiation process is the field emission phenomenon. To initiate a plasma in air at atmospheric pressure when microwaves interact with a metal wire, a main condition must be met: the power of the microwaves must be high enough to induce in wire a high electrical field. The magnitude of this electric field depends on several parameters: the length and diameter of the metal wire, its electrical conductivity, and the power of the microwaves. Therefore, the electrical voltage induced by microwaves in a metal wire can be summarized by two parameters: the ohmic resistance of the metal wire calculated for alternating current ($R_{AC}$) and the power of the microwaves in the focal point of the waveguide. Considering that the $R_{AC}$ of the titanium wire (5 cm length) calculated for the frequency of 2.45 GHz at 20 °C is 437 Ohm, the electrical voltage induced by microwaves in the titanium wire is 46.7 kV for 500 W microwave power and 59.2 kV for 800 W microwave power [27]. At a value of 59.2 kV, the breakdown of the air is initiated, and a plasma is formed. For lower values, the plasma is not initiated. Therefore, to initiate the plasma in air at atmospheric pressure, the power of the microwaves was increased to 800 W. After the plasma initiation process is started, the power of the microwaves can be decreased. From the last figure in Section 4 one can note that the plasma column generated at 500 W microwave power is much smaller than the plasma column generated at 800 W. In Figure 1 and Figure 2 one can observe that during microwave discharges obtained at 500 W and 800 W RF, both plasmas emit spectral lines corresponding to ions from gas and metal wire. To determine the plasma sustaining conditions under 800 W microwave power, the local thermodynamic equilibrium (LTE) conditions for both plasmas (500 W vs. 800 W microwave power) were evaluated. In Figure 3 and Figure 4 it can be observed that both plasmas have fulfilled the conditions of LTE. The plasma-initiating process is caused by the collision between the electrons emitted by the wire and the molecules of the air, in this process, a large amount of heat is generated. The process of electron–atom collision triggers the thermionic emission. After the thermionic emission occurs, the microwave power can be reduced. Therefore, the plasma sustaining process at microwave powers below 800 W is due to the initiating of the thermionic emission. If the microwave power is reduced below 500 W microwave power, the plasma is not initiated. During microwave discharges (500 W and 800 W microwave powers) a mixture of TiO and TiO_2_ nanoparticles was generated. From the analysis of Figure 5a,b, we noticed that, although there is a large amount of the TiO_2_ and TiO mixture at the tip of the titanium wire after exfoliation of the titanium oxide, the titanium wire has almost the same diameter. Therefore, during plasma generation the surface of the Ti wire was oxidized uniformly. Additionally, one can notice that at 500 W microwave power, the Ti wire did not vaporize in the focal point of the waveguide and Ti oxide was generated on the surface of the Ti wire.

At 800 W microwave power, the Ti wire was completely vaporized in the focal point region of the waveguide and a large number of nanoparticles of the Ti oxide were deposited on the waveguide walls. Given that in a microwave discharge a lot of heat is produced, the hot air flow induced by the plasma generated at 800 W microwave power carried away the oxide particles from the Ti wire towards the walls of the waveguide. At 500 W of microwave power, the vaporization of the Ti wire is significantly slower, and not enough heat was generated to form an airflow strong enough to carry the synthetized nanoparticles. In Figure 5a (1000× magnification) one can notice that the surface of the titanium oxide layer has an irregular shape with small bumps, which indicates that the Ti oxide has reached the boiling point. Due to the very high temperature generated by the plasma, the Ti oxide melted and then the liquid accumulated at the tip of the titanium wire. If one compares the EDS analysis performed on the two samples (silicon substrate deposited with TiO_x_ and Ti wire covered with TiO_x_ layer), it is observed that microwave power plays an important role in quantitative conversion from Ti wire to TiO_x_. By using this microwave plasma generation method, Ti oxide powders or Ti oxide coatings can be obtained on the surface of Ti wires depending on the microwave power. To oxidize a metal wire over a longer length than that observed during these experiments, once the microwave discharge has been initiated, the wire should be continuously advanced through the focal point of the waveguide.

The main advantages of the new microwave plasma method are high production efficiency, a complete conversion of the Ti wire into TiO_x_ nanoparticles, and synthesis of crystalline TO_x_ nanoparticles even if the atmospheric air contains water vapor [28,29] or other impurities. Moreover, the microwave device is a portable system (it weighs around 12 kg, the dimensions are 0.5 m length, 0.5 m width, and 0.2 m height), it can be installed anywhere, and it does not require special rules/authorizations for use as in the case of using lasers or storage of compressed gas cylinders. This very simple and inexpensive microwave device can be applied in the commercial field, in dental implant processing, or in the field of green energy. Due to the UV radiation reflection properties of Ti oxides [30], these powders can be used in the commercial domain as raw material in sunscreen manufacturing or other applications. Using this microwave plasma method for dental implant coating can improve their biocompatible and osseointegration properties. Considering that TiO_2_ is antimicrobial and that rough implant surfaces can increase bone-to-implant contact [31], the lifespan of an implant can be extended by depositing a layer of TiO_2_ nanoparticles on the implant surface. In the green energy domain, the microwave device can be used to manufacture TiO_2_ electrodes covered with noble metals [32] aiming to improve the photocatalysis system’s efficiency for green hydrogen production.

## 4. Materials and Methods

The evolution of technology has allowed for the development of microwave emission sources (magnetron tubes) for commercial applications. In the scientific field, different microwave plasma devices have been developed [33]. In general, a microwave plasma system is based on a few basic components: a microwave source, a power supply, and a resonant cavity or a waveguide. A new microwave plasma generator built with commercial components was developed by our group to create a plasma and synthesize titanium oxide films and nanoparticles. It is known that when a metallic object is introduced in a microwave oven, electric sparks are generated [34]. To create a plasma following the interaction between microwaves and a metallic wire the multimode waveguide of a microwave oven was replaced with a single-mode waveguide (TM_011_—transverse magnetic mode). The single-mode waveguide focuses the microwaves on a single point, which is the node of the cavity, where a high power density can be achieved.

The geometrical dimensions of the waveguide were calculated from Equation (1) [35]:(1)$$\left(f_{r}\right)\begin{array}{c} TM\\ mnl \end{array}=\left(\frac{1}{2\pi \sqrt{\mu \cdot \epsilon }}\right)=\sqrt{{\left(\frac{p_{01}}{a}\right)}^{2}+{\left(\frac{l\pi }{h}\right)}^{2}}$$
where $f_{r}$—the resonance frequency of the waveguide; a—radius of the waveguide, h—length of the waveguide; l—longitudinal mode of the cavity; μ—permeability of the medium within the cavity (H/m); ε—permittivity of the medium within the cavity (F/m); $P_{01}$—first zero of the Bessel function (equal to ~2.405).

The indices mnl of the TM propagation mode refer to the number of half-wavelength variations in the radial, axial, and longitudinal directions. From Equation (1), the waveguide has the following dimensions: 10.5 cm diameter, and 11 cm length. If a metallic wire, which is electrically isolated is placed with its tip in the focal point of the cavity, it will be irradiated by a high electric field when the power is applied. In atmospheric air, the high electric field generated at the tip of the metallic wire ionizes the gas atoms, and a plasma is created. The design of the new device is sketched in Figure 15.

A commercial microwave oven is usually powered at 220 V, 50 Hz. The magnetron generates microwaves with frequency = 2.45 GHz in pulsed mode (20 ms pulses or 50 Hz). To control the microwave power emission of the magnetron an electronic controller was developed. To analyze the behavior of the titanium wires when these were exposed to microwaves we used wires with a length of 5 cm from a roll of 99.98% purity titanium wire with 1 mm diameter. Table 3 displays the chemical composition of the Ti wire before exposure to microwaves and Figure 16 displays the EDS spectrum acquired from the Ti wire.

After positioning the titanium wire with one end in the focal point of the waveguide cavity, the power source of the magnetron was turned on, and then the power of the microwaves was increased to 800 W. After the plasma initiation process for the first Ti wire the microwaves power was set at 500 W and for the second Ti wire the microwaves power was set at 800 W. Following the interaction of the microwaves and the titanium wire, a plasma was generated.

The plasma generation process was set to 10 s for each experiment. In Figure 17a,b images of the plasma generated at 500 W and 800 W microwave power are displayed.

## 5. Conclusions

Using a direct interaction between microwaves and an electric isolated Ti wire in the air at normal atmospheric pressure a plasma was generated. The plasma volume increases with increasing microwave power. The LTE conditions were fulfilled for plasmas generated at 500 W and 800 W microwave power. During plasma generation, Ti oxide layers and nanoparticles were synthesized. The vaporization time of titanium wire and the nanoparticles deposition mechanism depends on the microwave power from the waveguide. During the 10 s interaction time of the Ti wire with 800 W power of the microwave, a 5 mm length of the Ti wire was completely converted to TiO_x_ nanoparticles.

The SEM and XRD analyses highlight that during microwave discharge at 500 W and 800 W microwave powers TiO_x_ nanoparticles with the same dimension were generated.

The EDS analysis performed on the two samples (silicon substrate deposited with TiO_x_ at 800 W microwave power and the growth of the TiO_x_ layer on the Ti wire at 500 W microwave power) showed that the microwave power plays an important role in quantitative conversion from Ti wire in TiO_x_.

The water vapor or other impurities contained in the atmospheric air do not affect the purity of the TiO_x_ nanoparticles generated during the interaction process. The TEM analysis indicates that most of the nanoparticles are either amorphous or nanocrystalline.

## Figures and Tables

**Figure 1 ijms-26-02173-f001:**
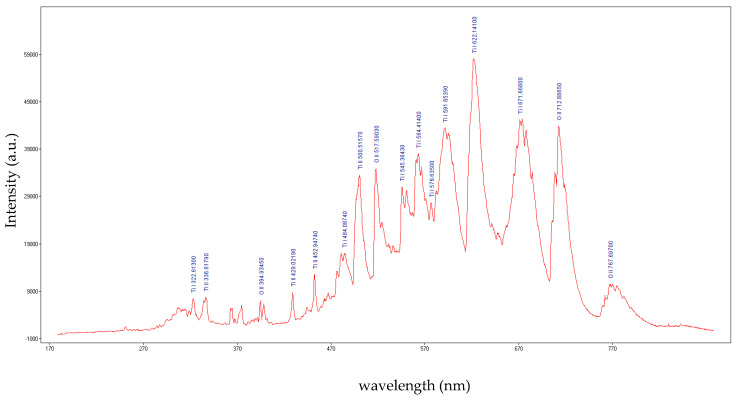
The optical emission spectrum of the plasma generated at 500 W microwave power.

**Figure 2 ijms-26-02173-f002:**
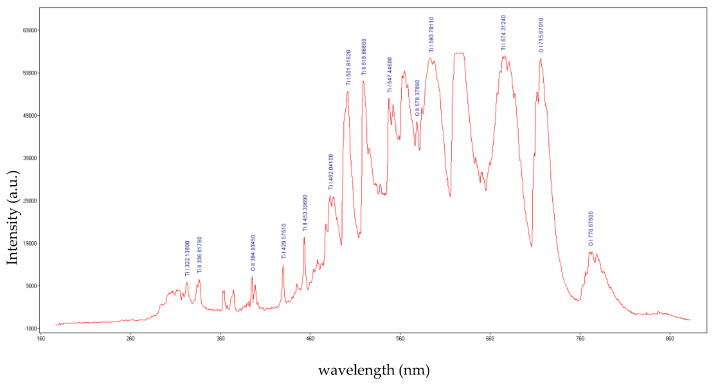
The optical emission spectrum of the plasma generated at 800 W microwave power.

**Figure 3 ijms-26-02173-f003:**
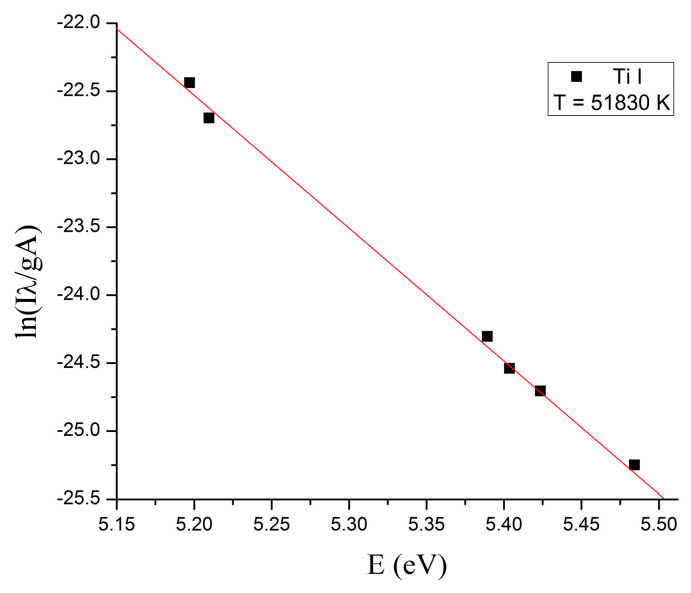
The Boltzmann plot of the plasma generated at 500 W microwave power.

**Figure 4 ijms-26-02173-f004:**
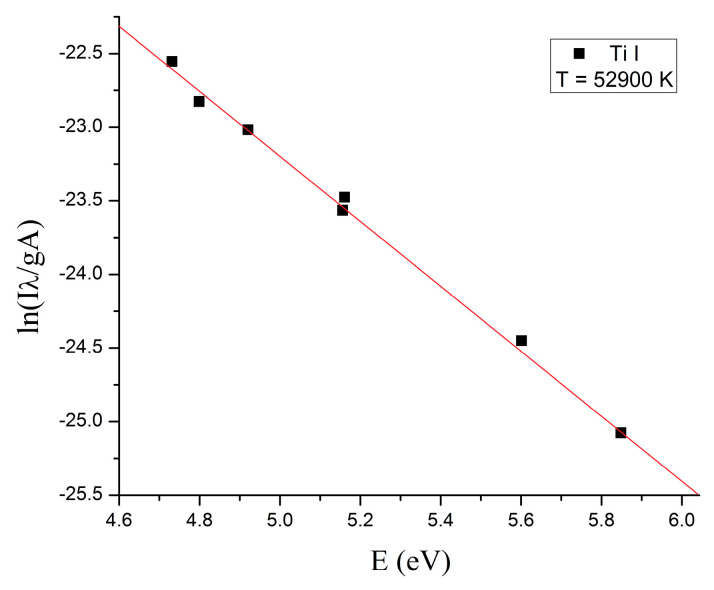
The Boltzmann plot of the plasma generated at 800 W microwave power.

**Figure 5 ijms-26-02173-f005:**
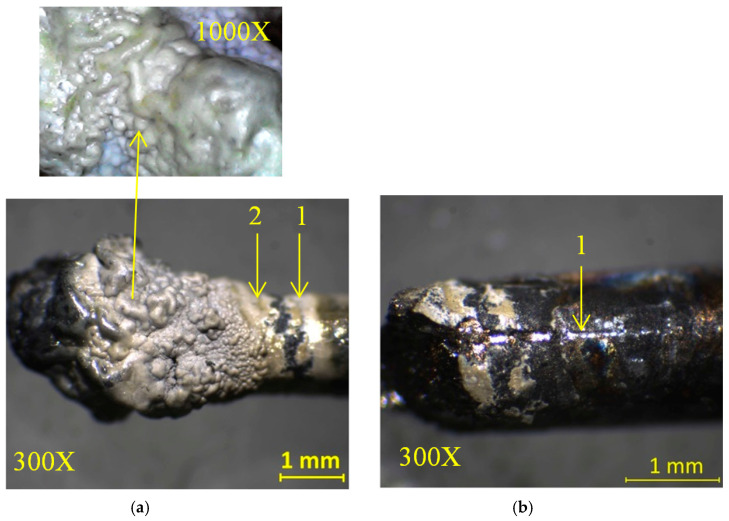
(**a**) Optical microscope images of the Ti wire after exposure to 500 W microwave power (magnification 300×): 1—the point where the needle for the adhesion testing instrument was fixed on TiO_2_; 2—the point where Ti oxide was exfoliated from the titanium wire. (**b**) Titanium wire after exfoliation of Ti oxide (magnification 300×), 1—the point where the instrument needle was fixed for the adhesion testing.

**Figure 6 ijms-26-02173-f006:**
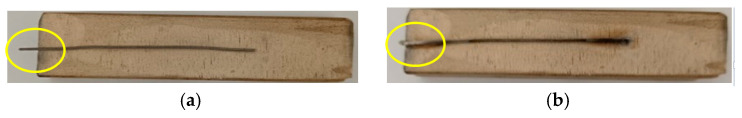
Images of the Ti wire before (**a**) and after exposure (**b**) to the microwave field.

**Figure 7 ijms-26-02173-f007:**
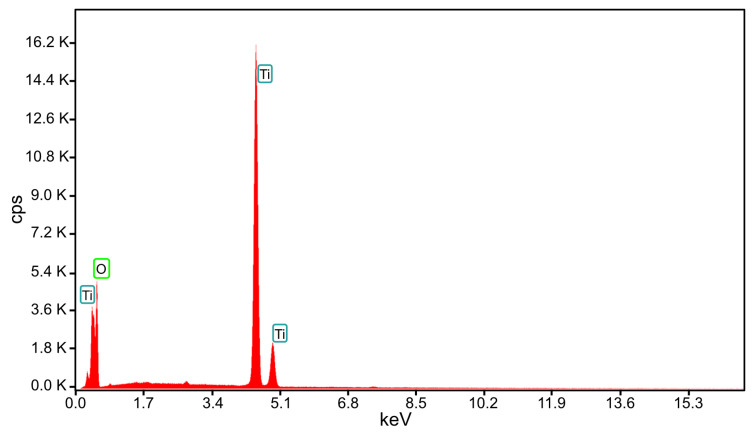
EDS spectrum of the TiO_x_ from the Ti wire surface.

**Figure 8 ijms-26-02173-f008:**
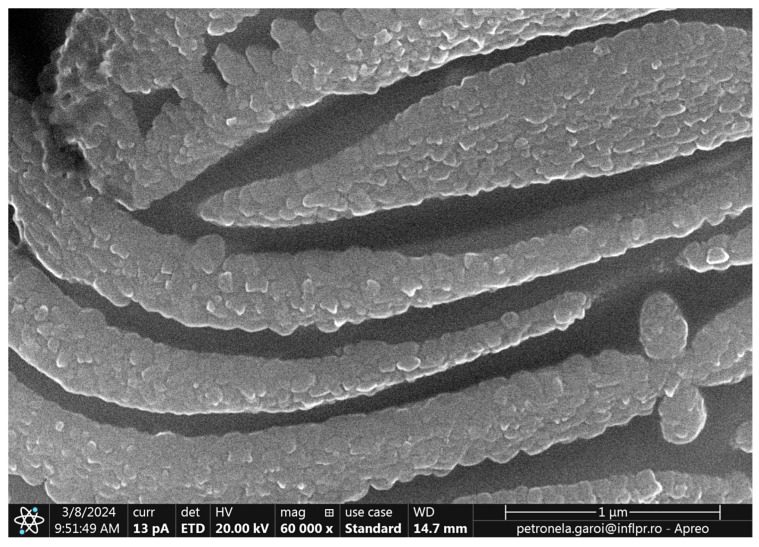
SEM image of the titanium wire irradiated with 500 W microwaves.

**Figure 9 ijms-26-02173-f009:**
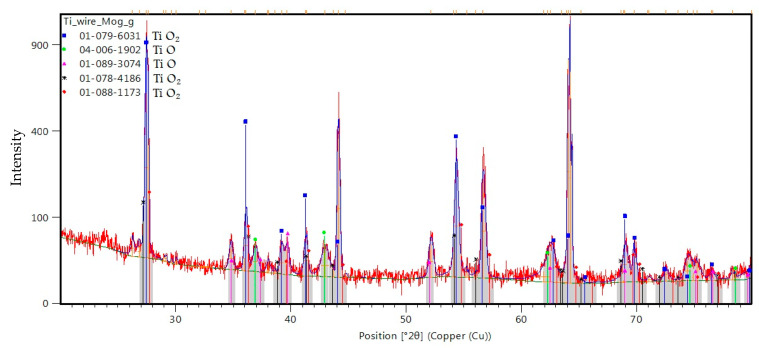
XRD analysis of the titanium oxide layer synthesized on the surface of the titanium wire by 500 W irradiation.

**Figure 10 ijms-26-02173-f010:**
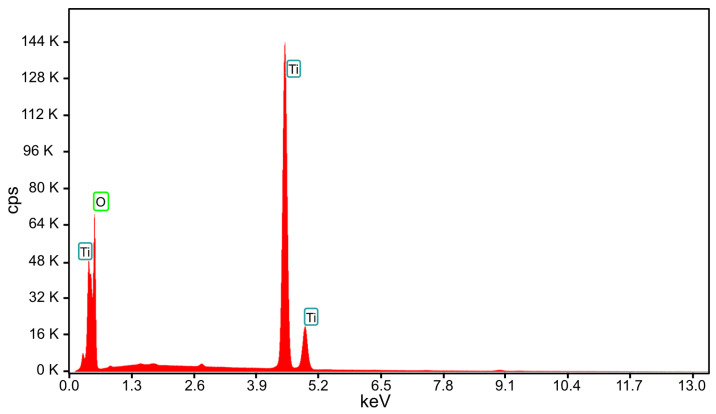
EDS spectrum of the TiO_x_ nanoparticles from the silicon substrate.

**Figure 11 ijms-26-02173-f011:**
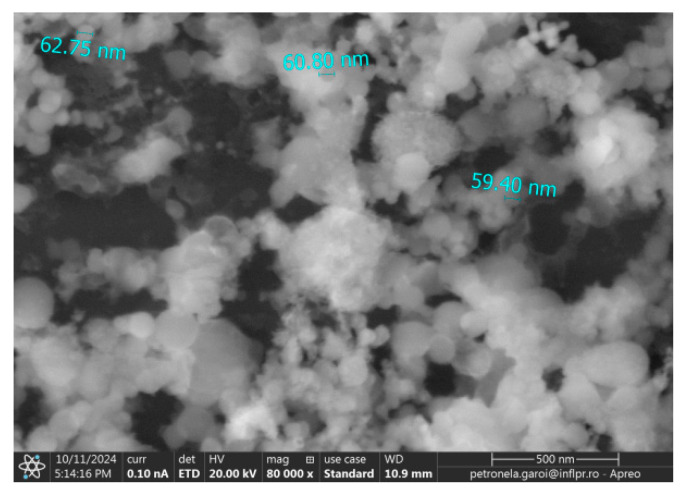
SEM image of the nanoparticles deposited on the silicon substrate.

**Figure 12 ijms-26-02173-f012:**
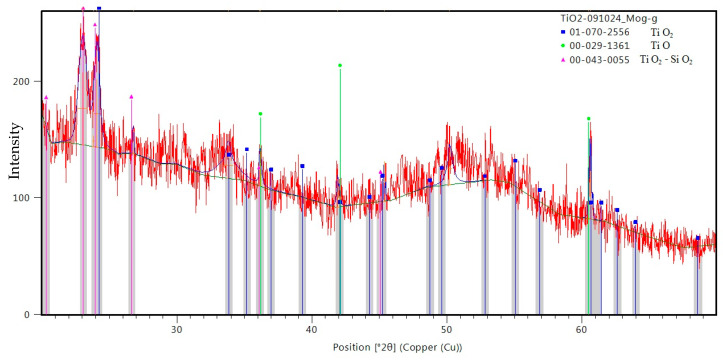
XRD analysis of the Ti oxide nanoparticles deposited on the Si support.

**Figure 13 ijms-26-02173-f013:**
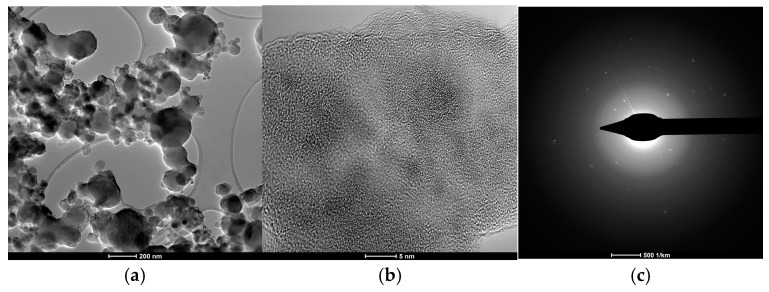
(**a**) Bright-field TEM image of the Ti oxides nanoparticles, (**b**) high-resolution TEM image of the nanoparticles, and (**c**) selected area electron diffraction pattern acquired from the nanoparticles.

**Figure 14 ijms-26-02173-f014:**
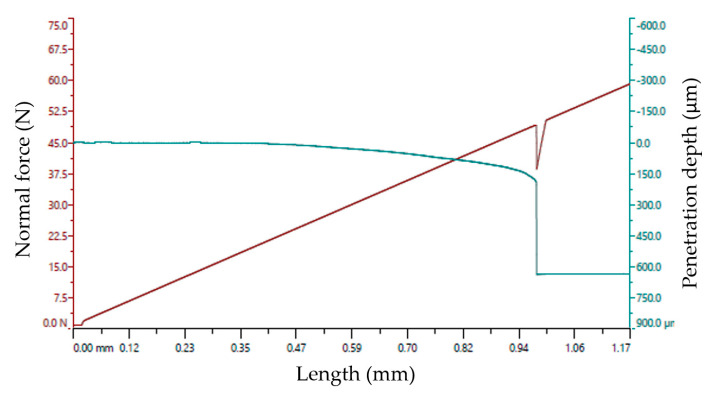
Adhesion analysis of TiO_x_ layer on the Ti wire.

**Figure 15 ijms-26-02173-f015:**
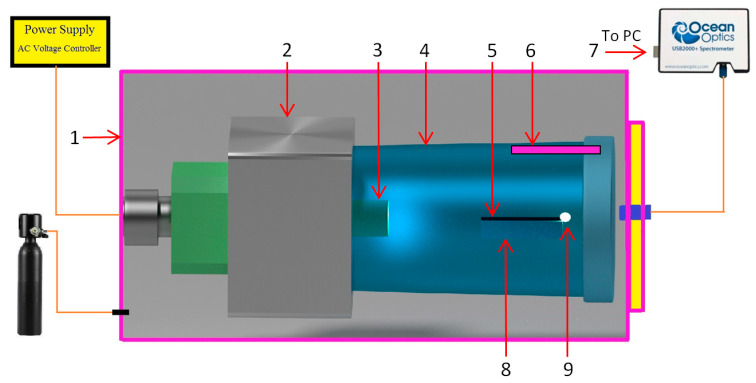
Design of the microwave plasma generator. 1—pressure chamber, 2—magnetron, 3—magnetron antenna, 4—TM waveguide, 5—metallic wire, 6—metallic vapors deposition substrate, 7—NIR-VIS-UV Spectrometer, 8—ceramic support, 9—plasma.

**Figure 16 ijms-26-02173-f016:**
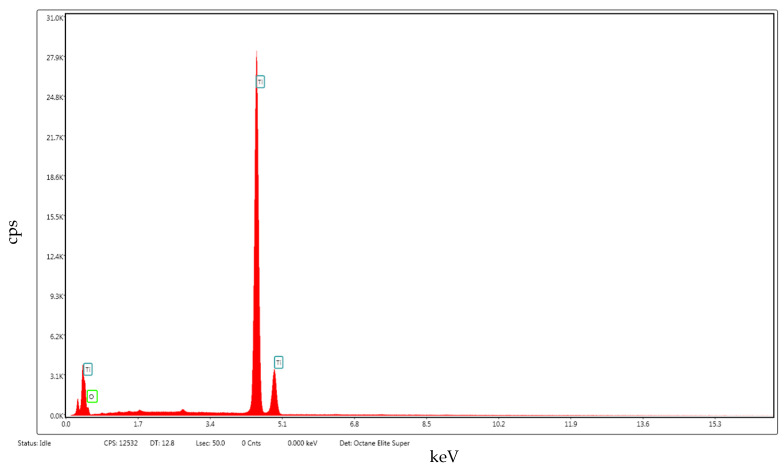
EDS spectrum from the Ti wire before exposure to microwaves.

**Figure 17 ijms-26-02173-f017:**
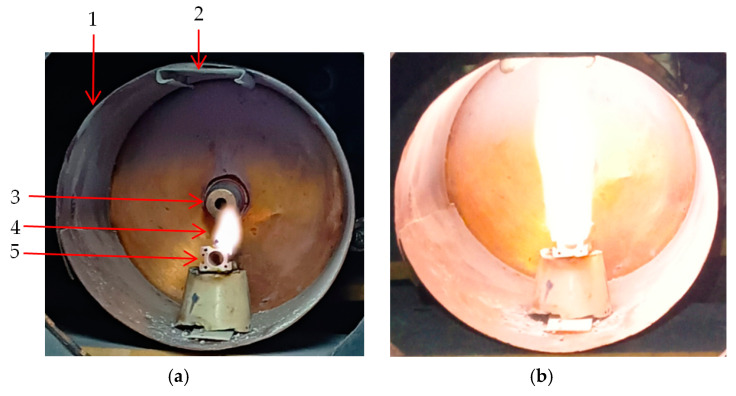
The microwaves discharge at 500 W RF (**a**) and 800 W RF (**b**) in air at atmospheric pressure: 1—waveguide, 2—the place where the silicon substrate was inserted, 3—magnetron antenna, 4—plasma, 5—ceramic support.

**Table 1 ijms-26-02173-t001:** The chemical composition of the TiO_x_ from the Ti wire surface.

Element	Weight %	Atomic %
O K	36.95	63.7
Ti K	63.05	36.3

**Table 2 ijms-26-02173-t002:** The chemical composition of the TiO_x_ nanoparticles synthesized at 800 W microwave power.

Element	Weight %	Atomic %
O K	46.14	71.95
Ti K	53.86	28.05

**Table 3 ijms-26-02173-t003:** The chemical composition of the Ti wire before exposure to microwaves.

Element	Weight %	Atomic %
O K	4.22	11.65
Ti K	95.78	88.35

## Data Availability

Data is contained within the article.

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
