# Peer review of "Synthesis of the Titanium Oxides Using a New Microwave Discharge Method"

_ijms, 2025, doi:10.3390/ijms26052173_

Round 1
Reviewer 1 Report
Comments and Suggestions for Authors
In this work, the authors reported a synthesis strategy of the titanium oxides using a new microwave discharge method. This benifits to boost this simple and inexpensive technique to grow a TiOx layer on the surface of titanium electrodes or can synthetized nanocrystalline TiOx particles. However, there is a big problem with the structure of the paper, for example, Figures 1 and 2 are placed at the end of the paper, and Figures 3 and 4 are placed at the front, which is obviously very confusing. In addition, the quantitative chemical composition analysis and the High-angle annular dark-field (HAADF) of the TEM elemental maps besides EDX elemental mapping image of the samples after exposure to 500 W and 800 W microwave power are lacking in this present manuscript. Therefore, I recommend that this study could be published but after the paper structure adjustment and the experimental data mentioned above must be completed.
Comments on the Quality of English LanguageEnglish expression is OK.
Author Response
Dear Reviewer 1
Thank you for reviewing the article "Synthesis of the titanium oxides using a new microwave discharge method". Below I send you the answers to your questions.
In this work, the authors reported a synthesis strategy of the titanium oxides using a new microwave discharge method. This benifits to boost this simple and inexpensive technique to grow a TiOx layer on the surface of titanium electrodes or can synthetized nanocrystalline TiOx particles.
Question 1: However, there is a big problem with the structure of the paper, for example, Figures 1 and 2 are placed at the end of the paper, and Figures 3 and 4 are placed at the front, which is obviously very confusing.
Our answer: The paper was corrected in accord with the International Journal of Molecular Sciences template.
Question 2: In addition, the quantitative chemical composition analysis and the High-angle annular dark-field (HAADF) of the TEM elemental maps besides EDX elemental mapping image of the samples after exposure to 500 W and 800 W microwave power are lacking in this present manuscript.
Therefore, I recommend that this study could be published but after the paper structure adjustment and the experimental data mentioned above must be completed.
Our answer: The EDS analysis were added (figures 7,10 and Tables 1 ,2)
Thank you for your consideration of this manuscript.
Sincerely,
Dr. George Mogildea

Reviewer 2 Report
Comments and Suggestions for Authors
In the manuscript, the authors reported an approach toward “Synthesis of the titanium oxides using a new microwave discharge method". Generally, the current work is well carried out but the authors should try to emphasize better the importance of the current manuscript in order to attract the readership of this journal. This manuscript can be published justified after the authors consider the following miner points.
· The common expression in abstract could be cleared as added one sentences in first to interest
· The abstract needs to be modified to eliminate vague sentences and replace with actual significant data and findings. Is it possible for the authors to include some numerical results in the abstract?
· The introduction can be improved by providing a more critical discussion of recent related literature. Discuss the shortcomings of previous work and the gaps, and how this work intends to fill those gaps.
Revised Introduction section based on the structure below:
1. 1st paragraph: Problem statement
2. 2nd paragraph: Current ongoing solution
3. 3rd paragraph: Proposed solution in this work.
4. 4th paragraph: Summarized the current research novelty and objective of this work.
· The quality of some figures is inferior and needs to be improved.
· .In Section 3, The discussion effectively interprets the findings but could delve deeper into the implications of Synthesis of the titanium oxides using a new microwave discharge method.
· In Section4, more details on the raw materials should be provided, such as the purity
- There are a lot of writing errors and grammar problems in the article. I hope the author can check them carefully and correct them.
Comments on the Quality of English Language
In the manuscript, the authors reported an approach toward “Synthesis of the titanium oxides using a new microwave discharge method". Generally, the current work is well carried out but the authors should try to emphasize better the importance of the current manuscript in order to attract the readership of this journal. This manuscript can be published justified after the authors consider the following miner points.
· The common expression in abstract could be cleared as added one sentences in first to interest
· The abstract needs to be modified to eliminate vague sentences and replace with actual significant data and findings. Is it possible for the authors to include some numerical results in the abstract?
· The introduction can be improved by providing a more critical discussion of recent related literature. Discuss the shortcomings of previous work and the gaps, and how this work intends to fill those gaps.
Revised Introduction section based on the structure below:
1. 1st paragraph: Problem statement
2. 2nd paragraph: Current ongoing solution
3. 3rd paragraph: Proposed solution in this work.
4. 4th paragraph: Summarized the current research novelty and objective of this work.
· The quality of some figures is inferior and needs to be improved.
· .In Section 3, The discussion effectively interprets the findings but could delve deeper into the implications of Synthesis of the titanium oxides using a new microwave discharge method.
· In Section4, more details on the raw materials should be provided, such as the purity
- There are a lot of writing errors and grammar problems in the article. I hope the author can check them carefully and correct them.
Author Response
Dear Reviewer 2
Thank you for reviewing the article "Synthesis of the titanium oxides using a new microwave discharge method". Below I send you the answers to your questions.
In the manuscript, the authors reported an approach toward “Synthesis of the titanium oxides using a new microwave discharge method". Generally, the current work is well carried out but the authors should try to emphasize better the importance of the current manuscript in order to attract the readership of this journal. This manuscript can be published justified after the authors consider the following miner points.
Question 1: The common expression in abstract could be cleared as added one sentences in first to interest. The abstract needs to be modified to eliminate vague sentences and replace with actual significant data and findings. Is it possible for the authors to include some numerical results in the abstract?
Our answer: The abstract has been modified in accordance with the requirements of the Reviewer 2.
Question 2: The introduction can be improved by providing a more critical discussion of recent related literature. Discuss the shortcomings of previous work and the gaps, and how this work intends to fill those gaps. Revised Introduction section based on the structure below:
- 1st paragraph: Problem statement
Our answer: Lines 58-67 “Current technologies based on plasma generation have low efficiency and incomplete conversion of the titanium to TiO2 [12,13]. Moreover, these systems contain mechanical and electronical modules that are very expensive, have large dimensions and cannot be easily installed in another workspace. The chemical techniques are less expensive, but they have limitations in terms of mass manufacturing, special condition requirements, as well as the use of toxic or poisonous chemicals that endanger human health and the environment. [14]. In addition, after manufacture of the nanoparticles it is necessary to implement procedures for washing and drying the nanoparticles. These procedures require time and money spent, which leads to a higher cost of producing nanoparticles.”
- 2nd paragraph: Current ongoing solution
Our answer: Lines : 54-56 , 67-69 “ Until now, TiO2 and TiO thin films were successfully synthesis by chemical and physical techniques such as chemical bath deposition (CBD) [7], chemical vapor deposition (CVD) [8], molecular beam epitaxy (MBE) [9], RF magnetron sputtering, electrodeposition [10] and pulsed laser deposition (PLD) [11]. Considering that the demand for TiOx nanoparticles and new materials based on TiOx is increasing, our study investigated the grow of TiOx directly on the surface of titanium wires and the synthesis of TiOx nanoparticles using a new microwave plasma method.”
- 3rd paragraph: Proposed solution in this work.
Our answer: Lines : 101-103 “Therefore, the microwave plasma method describes in this paper uses the same physical process to generate a plasma, which is oxidizing an electrical insulated titanium wire at low power and synthesis titanium oxides nanoparticles at high power levels.”
- 4th paragraph: Summarized the current research novelty and objective of this work.
Our answer: Lines 104-109 “Considering that numerous studies have been carried out recently on the interaction of metal powders/wires or small metal objects with microwaves of a certain power, this work highlights different behaviors of Ti wires under the action of low and high microwave power levels. Moreover, using this inexpensive method one can obtain a high efficiency conversion of the titanium in crystalline TOx powder nanoparticles and grow of the TOx layers on the surface of the Ti electrodes in atmospheric air.”
- The quality of some figures is inferior and needs to be improved.
Our answer: The quality of some figures has been improved (especially figure 14)
Question 3: In Section 3, The discussion effectively interprets the findings but could delve deeper into the implications of Synthesis of the titanium oxides using a new microwave discharge method.
Our answer: Lines : 365-371 “The main advantages of the new microwave plasma method are a: have high production efficiency, a complete conversion of the Ti wire in to TiOx nanoparticles and synthesis crystalline TOx nanoparticles even if the atmospheric air contains, water vapor [28, 29] or other impurities. Moreover, the microwave device is a portable system (it weighs around 12 kg, the dimensions are 0.5m length, 0.5m width and 0.2m height) , it can be installed anywhere, does not require special rules/authorizations for use as in the case of using lasers or storage of compressed gas cylinders.”
Question 4: In Section4, more details on the raw materials should be provided, such as the purity
Our answer: Lines: 420-424 “To analyze the behavior of the titanium wires when these were exposed to microwaves we used wires with a length of 5 cm from a roll of titanium wire 99.98% purity with 1 mm diameter.
In Table 3 is displayed the chemical composition of the Ti wire before exposure at microwaves and in figure 16 is displayed the EDS spectrum acquired from the Ti wire.”
Question 5: There are a lot of writing errors and grammar problems in the article. I hope the author can check them carefully and correct them.
Our answer: The quality of English language was improved.
Thank you for your consideration of this manuscript.
Sincerely,
Dr. George Mogildea

Reviewer 3 Report
Comments and Suggestions for Authors
The authors have reported on the synthesis of the titanium oxides using a new microwave discharge method.
I have the following comments on the manuscript.
- Please provide the full form of TiOx for first use in the abstract. It should be corrected as titanium oxides (TiOx).
- As the authors have given the Material and Method in section 4, the figure numbers are not in order. The Material Method section should be after the introduction and it should be section 2.
- There are no X and Y axis titles in figure 3, figure 4, figure 14. The X and Y axis titles as well as the inset of the figures should be written such that it is properly visible. (for example, the figure 5 and figure 6 are clearly visible).
The font size of the scale values for Figure 3 and Figure 4 is not visible at all. The font size should be larger in the figures.
- In page number 5, line 129, there are typographical mistakes. It should be “In figures 8 (a) and (b) images”.
- The value of the scale bar in the TEM images in Figure 13 is very small. Please provide the value of the scale bar in larger font.
- The conclusion part is too short. The authors should give a proper conclusion of the findings. The authors should provide the conclusion of the results from the SEM XRD, and TEM analysis.
- The figure caption should be uniformly written as “ Fig. 1, Fig.2, Fig. 3, Fig. 4 etc….”
- In figure caption 7, TiO2 should be corrected. 2 should be subscript. Same mistake in page 7 and reference 4.
- Authors should check the English of the manuscript thoroughly for suitability of publication.
Although the manuscript is technically suitable for publication, there are several errors/mistakes in the manuscript.
I recommend a major revision of the manuscript to further consider for publication.
Comments on the Quality of English LanguageAuthors should check the English of the manuscript thoroughly for suitability of publication.
Author Response
Dear Reviewer 3
Thank you for reviewing the article "Synthesis of the titanium oxides using a new microwave discharge method". Below I send you the answers to your questions.
The authors have reported on the synthesis of the titanium oxides using a new microwave discharge method. I have the following comments on the manuscript.
Question 1: Please provide the full form of TiOx for first use in the abstract. It should be corrected as titanium oxides (TiOx).
Our answer: The abstract has been corrected in accordance with the requirements of the Reviewer 3.
Question 2: As the authors have given the Material and Method in section 4, the figure numbers are not in order. The Material Method section should be after the introduction and it should be section 2.
Our answer: The paper was corrected in accord with the International Journal of Molecular Sciences template.
Question 3: There are no X and Y axis titles in figure 3, figure 4, figure 14.
Our answer: The X and Y axis titles of the figures were added.
Question 4: The X and Y axis titles as well as the inset of the figures should be written such that it is properly visible. (for example, the figure 5 and figure 6 are clearly visible).
The font size of the scale values for Figure 3 and Figure 4 is not visible at all. The font size should be larger in the figures. In page number 5, line 129, there are typographical mistakes. It should be “In figures 8 (a) and (b) images”. The value of the scale bar in the TEM images in Figure 13 is very small. Please provide the value of the scale bar in larger font.
Our answer: Some figures have been enlarged so that the scale of values ​​can be more easily observed,
Question 5: The conclusion part is too short. The authors should give a proper conclusion of the findings. The authors should provide the conclusion of the results from the SEM XRD, and TEM analysis.
Our answer: The conclusions of the paper were improved in accordance with the requirements of the Reviewer 3
Question 6: The figure caption should be uniformly written as “ Fig. 1, Fig.2, Fig. 3, Fig. 4 etc….”
In figure caption 7, TiO2 should be corrected. 2 should be subscript. Same mistake in page 7 and referenc Our answer: The text was corrected.
Question 7: Authors should check the English of the manuscript thoroughly for suitability of publication. Our answer: The quality of English language was improved.
Thank you for your consideration of this manuscript.
Sincerely,
Dr. George Mogildea

Round 2
Reviewer 1 Report
Comments and Suggestions for Authors
After closely inspecting the revised MS and the responses to the reviewer, I have noticed that the problems concerted by the reviewer have been addressed. Thus, I agree to accept the MS for publication in its current form.
Author Response
Dear Reviewer 1
Thank you for reviewing the article "Synthesis of the titanium oxides using a new microwave discharge method".
Sincerely,
Dr. George Mogildea

Reviewer 3 Report
Comments and Suggestions for Authors
The manuscript is revised based on the comments of the reviewer.
However there are more corrections needed as followings.
- Please give full form of SEM, EDS, TEM and Ti. The abstract should be a single paragraph.
- Please give better quality figures for figure 7 and figure 10. The X and Y axis values are not visible at all. Further, denoted peaks for element also not visible clearly.
Figure should be clear enough for final acceptance version of the manuscript, otherwise it is really confusing for readers to understand the content of the article.
Based of the above comments, I recommend minor revision of the manuscript.
Author Response
Dear Reviewer 3
Thank you for reviewing the article "Synthesis of the titanium oxides using a new microwave discharge method". Below I send you the answers to your questions.
The manuscript is revised based on the comments of the reviewer.
Question 1: However there are more corrections needed as followings.
Please give full form of SEM, EDS, TEM and Ti.
Our answer: I have uploaded a document containing the SEM, EDS, TEM and Ti images at a higher resolution.
Question 2: The abstract should be a single paragraph.
Our answer: The abstract has been corrected.
Question 3: Please give better quality figures for figure 7 and figure 10. The X and Y axis values are not visible at all. Further, denoted peaks for element also not visible clearly.
Our answer: Figures 7 and 10 have been corrected.
Thank you for your consideration of this manuscript.
Sincerely,
Dr. George Mogildea
